# Interleaving Text and Number Embeddings to Solve Mathemathics Problems

**Marvin Alberts**[1,2,3]     **Gianmarco Gabrieli**[1]     **Irina Espejo Morales**[1]
[1]IBM Research     [2]University of Zürich     [3] NCCR Catalysis
{marvin.alberts, irina.espejo.morales}@ibm.com
gga@zurich.ibm.com

## Abstract

Integrating text and numbers effectively is a crucial step towards enhancing Large Language Models (LLMs) capabilities in assisting in scientific tasks. While most current approaches rely on discrete tokenization of numbers, for instance, conversion to scientific notation or base 10-decomposition, a recent approach proposed a continuous numerical encoding as an inductive bias. In this paper, we build upon this approach by introducing more expressive numerical embeddings. Our method addresses key shortcomings, including the elimination of numerical artefacts and the ability to handle a wide range of magnitudes without clipping.

Our work presents two key contributions. First, we employ an MLP to assign distinct directions in the embedding space to different numbers. Our second contribution is the introduction of a routing layer that differentiates between numerical and text embeddings. We hypothesise that this combined approach enables the model to distinguish between text and number distributions while maintaining its capacity for arithmetic operations.

Using only a 45 M parameter encoder-decoder architecture our method achieves a $R^2$=0.9988 over a wide range of magnitude $(10^{-3}, 10^8)$. In addition, we empirically observe a reduction of the numerical artefacts and biases observed compared to the baselines.

## 1   Introduction

Transformer-based autoregressive language models have revolutionized machine learning over the past years [1, 2, 3]. Originally designed for neural machine translation, these models have found applications across diverse domains, including natural language processing, image analysis, and mathematical problem-solving [4, 5, 6, 7]. Despite their versatility in natural language tasks, accurate mathematical reasoning remains a challenge [8], especially when Chain-of-Thought prompting or a scratchpad are not employed [9]. The limits of the transformer architecture to perform mathematical computations and predictions from first principles have been extensively explored in the literature[8, 10, 11, 12, 13]. However, the majority of approaches leverage digit-by-digit, scientific notation or base 10 formats to encode and decode numbers [14]. This discretization of numbers leads to inherent prediction artefacts, especially for unstructured and non-synthetic numerical input data. For this reason, XVAL has proposed to introduce continuity as an inductive bias [15], to reduce the impact of discontinuous tokenization. Despite the improved performance in regression tasks and the reduction of artefacts, XVAL limits the magnitude by normalising the numerical inputs. In this work, we built upon XVAL incorporating continuity as an inductive bias, and pose the question: *Does increasing the expressivity of the numerical embeddings, by allowing each number to attend to its surrounding text, improve the performance on number prediction in mathematical questions?* For instance, it is intuitive that the number "2" should attend to words like "eggs" or "apple" and the number "0.0001" to words like "infinitesimal" or "fraction". The second question we aim to address: *Can a routing*

38th Conference on Neural Information Processing Systems (NeurIPS 2024).

*layer (as in Mixture-of-Experts) differentiating between text and numbers induce structure in the prediction of the model to improve the generation of numbers?*

**Related Work**

Regarding number tokenization, a large number of LLMs use variants of Byte Pair Encoding (BPE) [16] to tokenize numbers in the same way as text. As shown by [8] and [12] this introduces a limit to the number understanding of these models. Charton 10 demonstrates that transformer models can perform complex matrix operations and proposes a scheme tokenizing numbers as a series of tokens for sign, mantissa and exponent with varying levels of precision. As noted by [15], this approach does not take into consideration the continuous nature of numbers which is important in scientific domains, and instead, the authors introduce an architecture called XVAL which bypasses the need for number tokenization but not for number preprocessing and assigns a to each number a single numerical embedding modulated by the magnitude of the number itself. In the arithmetics applications [11] that concern this paper [14], the work by [17] shows that by adding positional encodings to digits it is possible to generalize out of the maximum training length of digits.

## 2   Our approach

In this work, we introduce Multimodal Decoding (MMD) which builds on the XVAL[15] architecture. XVAL embeds numbers by introducing a number token, *<num>*. The embedding of this number token is multiplied by the actual value of the number, effectively scaling the embedding of the number token. To decode numbers XVAL introduces two heads, one for tokens and one for numbers. During decoding if the *<num>*-token is predicted the model embedding is routed to the number head predicting a number.

Our architecture is similar. We also embed numbers separately but instead of scaling an embedding, we use an MLP allowing each number a distinct embedding direction learned by attending the surrounding text and as such providing more expressivity (see Figure 1 (a)). Another major change consists of the introduction of a routing layer inspired by Mixture-of-Experts (MoE) models [18]. Instead of relying on the text head to predict the *<num>*-token and then passing the model embedding to the number head, the routing layer predicts whether a given model embedding is routed to the text or number head (see Figure 1 (b)). This approach is motivated by the following insights: i) An embedding for a given number will be attending to the text that surrounds it, and different numbers will attend to different words; ii) The routing layer will force text and number embeddings

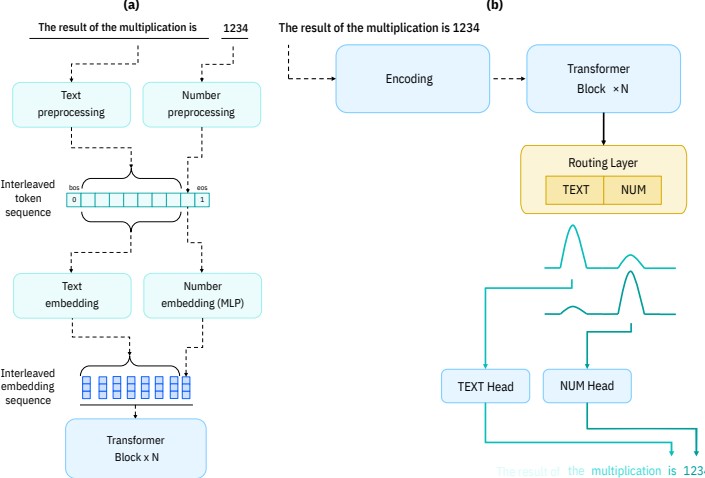

Figure 1: Diagram summarizing the MMD decoding method presented in this paper. (Left) High-level workflow of the encoder, routing layer classifying the modality, and the decoder. (Right) Zoom in on encoding an interleaved text and number sequence where, for text, the usual tokenization scheme is followed and for each number a new embedding vector is trained end-to-end.

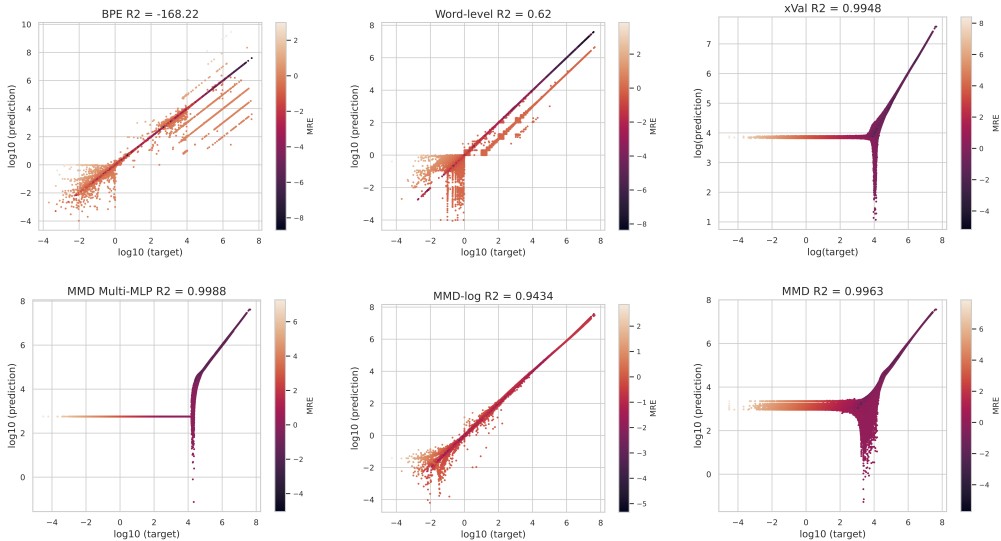

Figure 2: Comparison of log-log prediction vs ground truth values for arithmetic computations in the test set for baselines BPE and World-level (bottom row from left to right) and our MMD method (top row). The colour of each point indicates the relative error between prediction and ground truth with darker being a lower error.

to have a different distribution so that arithmetic calculations can be performed with numerical embeddings inside the transformer blocks; and iii) Instead of hard-coding an inductive bias for numerical embeddings with this method allows for finding optimal embedding distributions that might otherwise be missed.

## 3 Experiments

To evaluate the efficacy of our architecture, we conduct experiments on two progressively more complex tasks. We begin with a relatively straightforward problem: predicting numerical answers to arithmetic problems. In this task, we constrain the arithmetic problems such that all answers are numbers (*Numbers Only*).

We then extend our evaluation to more challenging math problems. In this case, we introduce problems that require either numerical or textual responses, or a combination of both (*Text and Numbers*).

Since we propose fundamental changes on numerical embeddings and different losses, we selected a simple dataset, the Mathematics dataset [19], see Appendix C. We use two versions of the dataset. One containing only numerical answers and the second containing prompts with both numbers as text as possible answers.

Table 1: Results for the numbers-only prediction experiments for baselines (BPE and Word-level) and for three versions of our MMD method.

|  | MAE $\downarrow$ | RMSE $\downarrow$ | MRE $\downarrow$ | $R^2$ $\uparrow$ |
|---|---|---|---|---|
| BPE | $8.28 \cdot 10^4$ | $1.05 \cdot 10^7$ | **0.123** | $-168.22$ |
| Word-level | $5.07 \cdot 10^4$ | $5.02 \cdot 10^6$ | $0.833$ | $0.616$ |
| XVAL | $1.13 \cdot 10^4$ | $4.51 \cdot 10^6$ | $6.04 \cdot 10^4$ | $0.9948$ |
| MMD (Ours) | $8.59 \cdot 10^3$ | $4.94 \cdot 10^5$ | $1.04 \cdot 10^4$ | $0.9962$ |
| MMD-log (Ours) | $2.22 \cdot 10^4$ | $1.93 \cdot 10^6$ | $0.143$ | $0.9434$ |
| Multi-MLP MMD (Ours) | $\mathbf{5.60 \cdot 10^3}$ | $\mathbf{2.86 \cdot 10^4}$ | $4.84 \cdot 10^3$ | **0.9988** |

Table 2: Results for the text and numbers prediction experiments for baselines (BPE and Word-level) and for three versions of our MMD method.

| | F1-SCORE ↑ | MAE ↓ | RMSE ↓ | MRE ↓ | $R^2$ ↑ |
|---|---|---|---|---|---|
| BPE | - | $2.41 \cdot 10^{15}$ | $3.92 \cdot 10^{17}$ | $4.02 \cdot 10^{14}$ | $-5.34 \cdot 10^{20}$ |
| Word-level | - | $4.94 \cdot 10^{11}$ | $8.12 \cdot 10^{13}$ | $3.52 \cdot 10^{8}$ | $-2.32 \cdot 10^{14}$ |
| XVAL | 0.764 | $4.98 \cdot 10^{6}$ | $3.59 \cdot 10^{7}$ | $6.72 \cdot 10^{7}$ | 0.141 |
| MMD (Ours) | 0.766 | $1.41 \cdot 10^{7}$ | $2.8 \cdot 10^{7}$ | $5.85 \cdot 10^{6}$ | **0.679** |
| MMD-log (Ours) | **0.880** | $\mathbf{4.72 \cdot 10^{5}}$ | $\mathbf{6.00 \cdot 10^{6}}$ | $\mathbf{7.33 \cdot 10^{3}}$ | $-6.70$ |
| Multi-MLP MMD (Ours) | 0.769 | $1.78 \cdot 10^{7}$ | $4.51 \cdot 10^{7}$ | $1.53 \cdot 10^{7}$ | $-0.230$ |

**Numbers Only**

We train a total of three variants of our model. The first with a one-layer MLP to encode numbers and similarly a one-layer MLP as the number head. In the second variant, we predict the log-transformed numbers and in the third variant, the number head consists of a three-layered MLP. We compare these models to three baselines: One transformer model trained with a BPE tokenizer, i.e. numbers are tokenised digit by digit, a word level tokenizer and lastly XVAL. All three of the variants of the model outperform the baseline on MAE, RMSE and $R^2$ (see Table 1). Especially encouraging is analysing the log-log plot of the predictions vs the targets (see Figure 2). Here we observe specifically for BPE and word-level artefacts with the model predicting multiple magnitudes off the target. On the other hand, we also observe failure modes in our architecture. While the predictions and targets track well starting from around $10^2$, the model fails to predict small values. This problem is addressed by log-transforming the numbers yielding a log-log curve closely following the identity.

**Text and Numbers**

We then extend the evaluation of our method to mathematical problems beyond pure arithmetic, encompassing tasks which comprise interleaved text and numbers in various problems (e.g., algebra). We leverage the same methods and baselines reported in Section 3 and report the results in Table 2. In this case, our methods obtain performance metrics that are several orders of magnitude better than the baselines, demonstrating the challenge of text-only models to process interleaved sequences. In particular, log-transformed numerical representations result in improved accuracy compared to the other two variants, with the only expection of the $R^2$ score for the base MMD version. Moreover, the results show that logarithmic pre-processing is pivotal for differentiating text and numbers, with an F1-score on the modality classification that is 0.11 higher than the other methods. As expected, the results are generally worse than in Section 3, due to the higher complexity of the interleaved multi-task dataset.

## 4 Conclusion

In this paper we investigated two strategies to interleave text and numbers as input to a standard encoder-decoder transformer architecture. The first strategy allows more expressive numerical embeddings by assigning a distinct vector space to different numbers. The second strategy introduces a routing layer that differentiates between text and number tokens. This serves to create a desirable structure in the embedding space differentiating between text and numbers without explicitly introducing any inductive bias. From our experiments on predicting the numerical outcomes of arithmetic computations, we observed that our approach exhibited the most balanced set of metrics and when comparing predicted vs true values, our method was the closest to the identity showing consistent performance along magnitudes ranging from $10^{-3}$ to $10^{8}$. We also note that even if a baseline performed well for a particular metric, the true vs predicted plots reveal artifacts and undesirable biases. The main limitation of this study is the simplicity of the arithmetic's dataset, which we intend to address in future work by incorporating more complex datasets and perform experiments against a wider set of numerical encodings and architectures in the literature.

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

# A Illustration of the routing layer

See Figure 3 for a graphical description of the different baselines and our methods. All experiments start with the same question and answer pair but text and numbers processed differently. For the baselines BPE and Word-level in yellow, the routing layer is inactive because numbers are processed as text only.

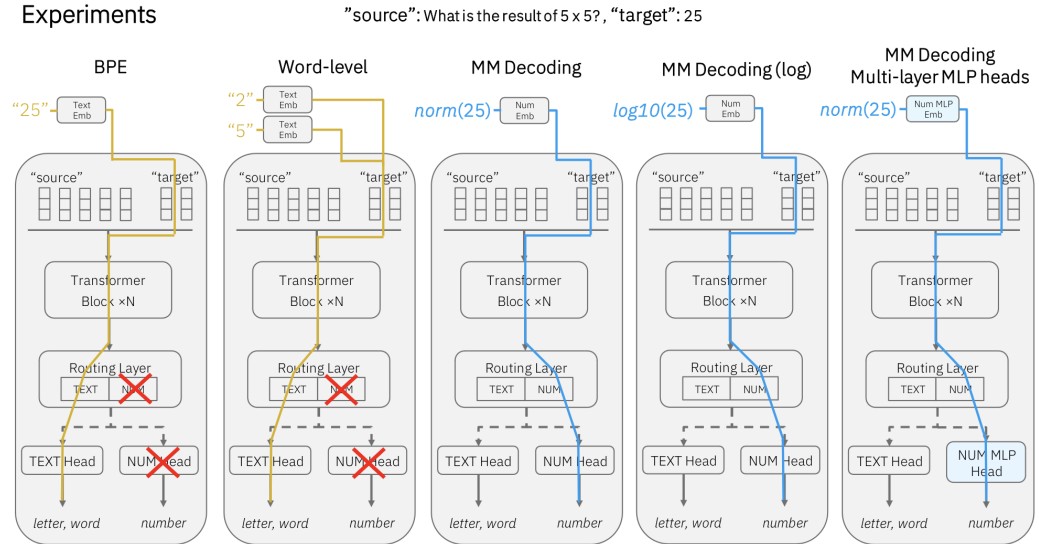

Figure 3: Schematic representation for the different baselines and our models indicating wether the routing layer is active or not when a number is in the input.

# B Experiment settings

**Model:** The backbone structure is a standard encoder-decoder architecture with 4 layers in the encoder transformer block and 4 layers for the text decoder block and a 2-layer fully-connected MLP for the numeric head. The activation functions are Gaussian Error Linear Units (GELU). All experiments were performed using a single NVIDIA A100 40GB GPU. Models were trained for 50 epochs by setting the learning rate to $1e-4$ for all the experiments.

**Metrics:** The metrics we use to evaluate number predictions for the result of arithmetic calculations are Mean Absolute Error (MAE), Mean Squared Error (MSE), Mean Relative Error (MRE), Median Relative Error (MedRE) and the $R^2$ coefficient. Instead of reporting one of them, together they present a better picture of the distribution and biases of errors with respect to the range of the true answer.

# C Data samples

Examples of question and answer pairs of the Mathematics dataset [19] that were used for training of *Numbers Only* and *Text and Numbers* experiments presented in the paper.

Question: Calculate -971810940.335 + 612120.
Answer: -970586700.335

Question: Solve -12*t - 4482 = 64*t - 383*t + 141*t for t.
Answer: 27

Question: Does 15 divide 8287819?
Answer: False

