# OpenReview forum: "Interleaving Text and Number Embeddings to Solve Mathemathics Problems"
_NeurIPS.cc/2024/Workshop/MATH-AI — MATH-AI 24_

### Official Review · Reviewer_idwq · 2024-10-07
**Approach aims to address limitations of XVal but the value of technique isn't evident from paper.**

**Rating:** 3
**Confidence:** 3

**Review:**

**Summary:** The paper presents a technique called Multimodal Decoding (MMD) which extends the numerical encoding technique from [XVal](https://arxiv.org/abs/2310.02989). MMD uses an MLP to encode the number and at decoding time adds a lightweight routing layer that acts as a soft classifier on the transformer representation and routes the embedding between text / numeric head. The numeric head directly predicts a number given an internal representation. Unlike the previous approaches that used a decoder-only architecture, the paper explores their technique on encoder-decoder architectures.

**Strengths:**
* **More Expressive Variant of XVal:** XVal represents numbers using a dedicated <num> token with input representation being <num> token scaled by number value and output representation being inferred via a numerical head for any <num> tokens generated in output. The current paper’s MMD approach makes this approach more flexible in two ways. First, it encodes the number using an MLP (with optional log application) rather than a single scaled embedding vector. Second, it gets rid of the intermediate token generation stage in decoding and instead uses a routing layer to optionally but directly activate the numerical head on intermediate representation.

**Weaknesses:**
* **Encoder-Decoder Architecture choice doesn’t seem well motivated:** Standard LLMs including XVal that this paper extends use decoder-only architecture. This paper uses Encoder-Decoder architecture (motivation isn’t specified) which prevents a cleaner comparison of the numerical embedding technique and limits the applicability of this paper.
* **Lacks Details:** Multiple relevant details seem missing from the paper e.g. a) the specifics of routing layer b) the specifics of the numeric and text-numeric dataset like size, train/eval splits etc c) the loss functions for which numerical head and the routing layer optimizes.
* **Technical Contribution seems Incremental:** The paper’s contributions w.r.t XVal while well-motivated are still quite incremental. The paper doesn’t address the key limitations of XVal. There is no comparison against open-source LLMs even at small scale (this was a limitation of XVal) which is crucial because despite the perceived issues in current digit based tokenization, it is undeniable that models using these tokenization techniques do fairly well on basic and even competitive math tasks. Which highlights that tokenization schemes cannot be evaluated in a vacuum and instead need to consider both the size of the transformer and the scale / diversity of the training input. XVal however highlighted their focus on continuous / large scale computations with focus on scientific datasets (something that LLMs do struggle with) but this paper doesn’t evaluate on those datasets. *So the value of MMD is not clear.*
* **Experiments do not seem well motivated:** The MMD technique is evaluated on a subset of data from [Saxton et al](https://arxiv.org/pdf/1904.01557) (which contains basic school level math problems). As also highlighted in the previous point, the value of these experiments is not really clear as today’s LLMs even at small scales like [Llama 1B and 3B](https://ai.meta.com/blog/llama-3-2-connect-2024-vision-edge-mobile-devices) already excel at harder version of these problems (e.g. [Hendrycks MATH](https://arxiv.org/abs/2103.03874)) despite the classic digit level tokenization.

**Questions:**
* **Is Numerical head trained using MSE ?** It wasn’t clear from the paper how the numerical head is trained. Does it use MSE similar to XVal ? Also do you have any explicit losses for training your routing layer ?
* **Routing Layer Specifics:** Do you have any explicit losses for training your routing layer ? What are the architectural specifics of the routing layer (is it like a one layer binary classifier ?)
* **Input Normalization:** Besides applying log in the log-MMD variant, do you have any other normalizations on your numerical inputs (within the network or in the training datasets). How do you handle numbers of large magnitude ?
* **Why Encoder-Decoder Architecture:** XVal used GPT-2 and other standard LLM use the decoder-only architecture. Is there some specific reason, your technique uses encoder-decoder ?
* **MAE / RMSE for Text-Numeric Dataset:** What do these metrics mean for predictions with both text and numbers ? Are you only computing it on numbers ?
* **Variance across reruns:** Given that MAE / RMSE are quite sensitive to magnitude of answer, did you observe high levels of variance across reruns of your approach (and if operating in log space reduced it) ?

**Suggested Corrections:**
* **Corrections to Figure 2:** Correct WorldLevel -> WordLevel. Baselines are in the top row and MMD is in the bottom row, caption says the opposite.
* **Rendering Issues:** Not sure if there are specific image formats used in this pdf but for some reason this paper wasn’t rendering properly for me with the reader lagging a lot in scrolling (tried multiple browsers, Preview, Adobe and to no avail). The issue was specific to this pdf. Wanted to highlight in case this issue was observed by / other reviewers. Apologies in advance if I may have missed any key details in my review above.

---

### Official Review · Reviewer_kXAM · 2024-10-07
**New multimodal model structure achieving better results on simple arithmetic dataset**

**Rating:** 6
**Confidence:** 3

**Review:**

The paper proposed a multimodal model, MMD, based on the XVal model, to solve the arithmetic problems. The two new strategies are: (1) individual embedding for each number and (2) an additional routing layer. The MMD model achieved better result for the Mathematic Dataset. The paper is well written with sound logic. The authors noted that the dataset used is simple and planned to test on more complex dataset in the future. Besides testing different dataset, I think the paper can benefit from:
(1) Experimenting on only the embedding change and only the routing layer - is the improvement coming more from one of them or is it a combined effect?
(2) Explaining why Multi-MLP MMD is better for numbers only problems and MMD-Log is better for text-and-numbers problems
(3) Listing details on model structure and training process such as embedding size to increase reproducibility
(4) Discussing computation trade-off - does adding a routing layer increase training time? does giving each number its own embedding increased overall model size?
(5) Providing sampled example where the XVal result is not as accurate as MMD

---

### Official Review · Reviewer_PFy7 · 2024-10-07
**Solve Math Problems by interleaving text and number embeddings**

**Rating:** 7
**Confidence:** 3

**Review:**

Summary:
The paper addresses the challenge of integrating text and numerical data in large language models (LLMs). The paper builds on recent advancements that propose continuous numerical encoding as a more effective way to incorporate numbers into LLMs. The authors employ MLP to assign distinct directions in the embedding space to different numbers and propose a routing layer that differentiates between numerical and text embeddings.

Quality:
The paper is technically sound.

Clarity:
The paper is well-written and easy to follow.

Originality:
This paper introduces Multimodal Decoding (MMD) which builds on the XVAL. I think the authors make good claims in the method section. The illustration of the routing layer at the end of the paper is good.

Pros:
- Proposed two contributions that resulting in an improved performance in the evaluation phase regarding F1-SCORE, MAE, RMSE, R^2 compared with other methods.

Cons:
- it would be helpful to include significant tests when comparing with other methods.

---

### Decision · Program_Chairs · 2024-10-08

Accept